# Social connections are differentially related to subjective age and physiological age acceleration amongst older adults

Daisy Fancourt [1] ✉, Andrew Steptoe [1] & Mikaela Bloomberg[2]

Human social connections are complex ecosystems formed of structural, functional and quality components. Weak social connections are associated with adverse age-related health outcomes, but we know little about the ageing-related processes underlying this. Using data from 7047 adults aged 50+ in the English Longitudinal Study of Ageing, we explore associations between diverse aspects of social connections and both older subjective age and accelerated physiological age using a validated physiological ageing combining cardiovascular, respiratory, haematologic and metabolic indicators. Doubly robust estimations using inverse-probability-weighted regression adjustment estimators show that living alone, low social integration and low social support are risk factors for physiological age acceleration. However, weak social connections did not have a statistically significant association with older subjective age. Analyses are robust to multiple sensitivity analyses and maintained four years later. We propose the hypothesis that accelerated physiological ageing may be a mechanism underpinning the relationship between weak social connections and age-related morbitidy and mortality.

Human social connections are complex ecosystems formed of structural components (such as the size of social networks and frequency of social contact), functional components (including the social and emotional support provided by these networks and contacts) and quality components (including the positive and negative experiences our interactions bring)[1]. Multi-national research demonstrates that social connections change over the life-course, with individuals prioritising certain types of social goals at different life points, such as focusing more on quality rather than quantity of social connections as they age. However, there are certain patterns that are well-evidenced and recognised as indicative not just of adaptive of compensatory patterns but instead of weak social connections that can be detrimental to health[2]. For example, estimates of the relationship between loneliness and age show U-shaped curves, with loneliness being higher amongst younger people but also steadily increasing from around age 60 onwards[3,4]. Similarly, social isolation becomes more prominent as people age, with severe social isolation being 4 times more likely in those aged >90 compared to aged 65–69[5].

These weak social connections are clearly and strongly associated with adverse health outcomes, including the incidence of physical diseases (e.g. cardiovascular disease/dementia/diabetes)[6–9], psychiatric disorders (e.g. depression/anxiety/ schizophrenia)[10,11], age-related decline[12,13], and mortality (through suicide and other causes)[14–16].

Recent biological research has identified a number of mechanistic pathways that link weak social connections to these age-related outcomes, including (i) an overactive HPA axis resulting in receptor cells developing glucocorticoid resistance, leading to a greater susceptibility to inflammation, (ii) dysregulation of the autonomic nervous system's ability to regular cardiovascular activity, (iii) maladaptive changes in immunological responses, (iv) reduced repair and restorative processes, and (v) increased brain atrophy and reduced neurogenesis[17–19]. These biological processes are notable because they themselves include important hallmarks of ageing, which do not necessarily advance in tandem with chronological age[20]. These processes have largely been looked at in isolation, with studies focusing on

[1]Department of Behavioural Science and Health, UCL, London, UK. [2]Department of Epidemiology and Public Health, UCL, London, UK. ✉e-mail: d.fancourt@ucl.ac.uk

individual biomarkers[21–24], but research into physiological ageing indices is providing alternative avenues for exploration. Physiological ageing indices combine phenotypic measures of age-related blood-based biomarkers (e.g. fibrinogen, C-reactive protein and glycated haemoglobin) with tests of broader age-related physiological function (e.g. grip strength, respiratory function, pulse pressure and blood pressure)[25–27]. Acceleration of physiological age relative to chronological age has been shown to predict multiple age-related outcomes including cardiovascular disease, arthritis, osteoporosis, cognitive dysfunction, dementia and mortality[25–27]. However, to date, there is little research into whether weak social connections relate to lower physiological age using such indices.

In addition to influencing physiological processes of ageing, it is also plausible that cultural engagement helps individuals to maintain a feeling of youthfulness, reducing their perceived or subjective age. Subjective age (asking an individual how old they feel) is a simple assessment of a complex phenomenon. How old someone feels is socially and culturally laden, reflecting factors such as social norms and personal desires. But it also reflects psychological and physiological processes, and these processes are related to maintaining good health and physical function. Psychologically, these processes include feelings of vitality, wellbeing, optimism and self-esteem[28,29], all of which are themselves related to lower risk of age-related morbidity and mortality[30]. Physiologically, these processes include a person's sense of their own functioning and capacity. Notably, subjective age can be an important marker in its own right: people who feel younger than they are chronologically have lower inflammation, accelerated epigenetic ageing, better preservation of cognition, reduced risk of future dementia, fewer future hospitalisations, and lower mortality risk, even after controlling for existing health status[28,31–37]. But whereas there is a strong literature on how subjective age relates to age-related morbidity and mortality, there is much less research into factors associated with an older subjective age, such as weak social connections.

Consequently, subjective and physiological age provide different but complementary insight into mechanistic processes that could link weak social connections to age-related morbidity and mortality. In this paper, we take a first step towards testing such a mechanism by exploring whether there are associations between weak social connections and older subjective age and accelerated physiological age. Assessing both subjective and physiological age in parallel is important as it provides the opportunity to explore whether correlations with weak social connections may be stronger for subjective or objective markers. We used a theoretically-informed model of social connections that takes account of their structural, functional and quality factors and made use of a representative cohort of older adults living in England[21]. Our research questions were (i) are strong social connections related to decelerated ageing, either subjective or physiological? (ii) how do potential effects differ based on structural, functional and quality aspects of social relationships? and (ii) is there synergy in the relationship between social connections and rates of subjective and physiological ageing? Here we show that structural factors such as living alone and low social integration as well as the functional factor of low social support are risk factors for physiological age acceleration. However, weak social connections are not related in a statistically significant way to older subjective age.

## Results
### Descriptive statistics
We used data from a large-scale panel study of adults aged 50 and over living in England; the English Longitudinal Study of Ageing (ELSA). Of the 7047 adults with relevant social and physiological variables eligible for inclusion in the analyses, the average age was 65.6 years (SD 9.3) and 55.2% were female.

We used a rich battery of questions on diverse aspects of structural, function and quality-related aspects of social connections. There were only weak to moderate correlations between measures of social connections, supporting their inclusion as separate items within the analyses (Fig. 2a and Supplementary Fig. 1), although people who had clear deficits in structural social factors like living alone scored lower on functional aspects like having lower social support and higher loneliness, and the quality of social relationships was generally poorer amongst those with poorer functional aspects too.

Subjective age was derived by asking participants how old they felt (measured in years) and subtracting chronological age from this, with negative scores indicating older subjective age, and positive scores indicating older subjective age. For physiological age acceleration, we used a previously-validated measure combining clinical indicators pertaining to the cardiovascular, respiratory, haematological and metabolic and musculoskeletal systems (Fig. 1). Negative scores indicate decelerated ageing (slower physiological ageing), and positive scores indicate accelerated ageing (faster physiological ageing). Notably, there was no statistically significant correlation between subjective and physiological age (r = −0.01, p = 0.56) (histogram and heatmap shown in Figs. 2c and 1d). People almost exclusively rated their subjective age to be the same or lower than their chronological age (92% the same or below), while physiological age was more evenly distributed above and below chronological age, with a slight skew towards people being older physiologically than chronologically (33% the same or below, Fig. 2b). There was a slight correlation between older chronological age and younger subjective age and a marked correlation between older chronological age and greater physiological age acceleration (scatterplots in Fig. 2e, f and hexagon heatmaps in Supplementary Fig. 2).

### Main analyses
Sample characteristics are shown in Table 1. For our main analyses, we calculated average treatment effects (ATEs) from doubly robust estimation using the inverse-probability-weighted regression adjustment (IPWRA) estimator. We adjusted for demographic, socio-economic, health and behavioural confounders within models, carrying out multiple sensitivity analyses to consider robustness to various model assumptions.

**Structure.** People who lived alone perceived their age to be 1.4 years younger than it was (95% CI 0.7 to 2.2 years younger), which was 16% lower than people who live with others (for full results see Fig. 3 & Table 2). But their physiological age was 2.6 years older (95% CI 1.4 to 3.8 years older; 50% greater age acceleration than people who live with others). People who had low levels of social integration also did not have a statistically significant older subjective age, although they did show patterns of accelerated physiological age (1.9 years older, 95% CI 0.8 to 3.0 years older, 37% greater age acceleration than those who were more socially integrated).

Amongst more tentative findings, people who were most socially isolated perceived themselves be 0.9 years younger than their chronological age (95% CI 0.2 to 1.7 years younger; 10% greater age deceleration than those not socially isolated; although this did not survive correction for multiple comparisons), whereas their physiological age was not materially different statistically. Having a small social network was also associated with being 0.9 years older (95% CI 0.04 to 1.8; 17% greater age acceleration; although this did not survive correction for multiple comparisons), but there was no statistically significant difference in subjective age.

**Function.** People who had low levels of social support or high levels of loneliness showed no statistically significant differences in their subjective age. But low social support was associated with physiological age acceleration (average 1.9 years older, 95%CI 0.9 to 2.8 years older, 35% greater age acceleration than those without low social support).

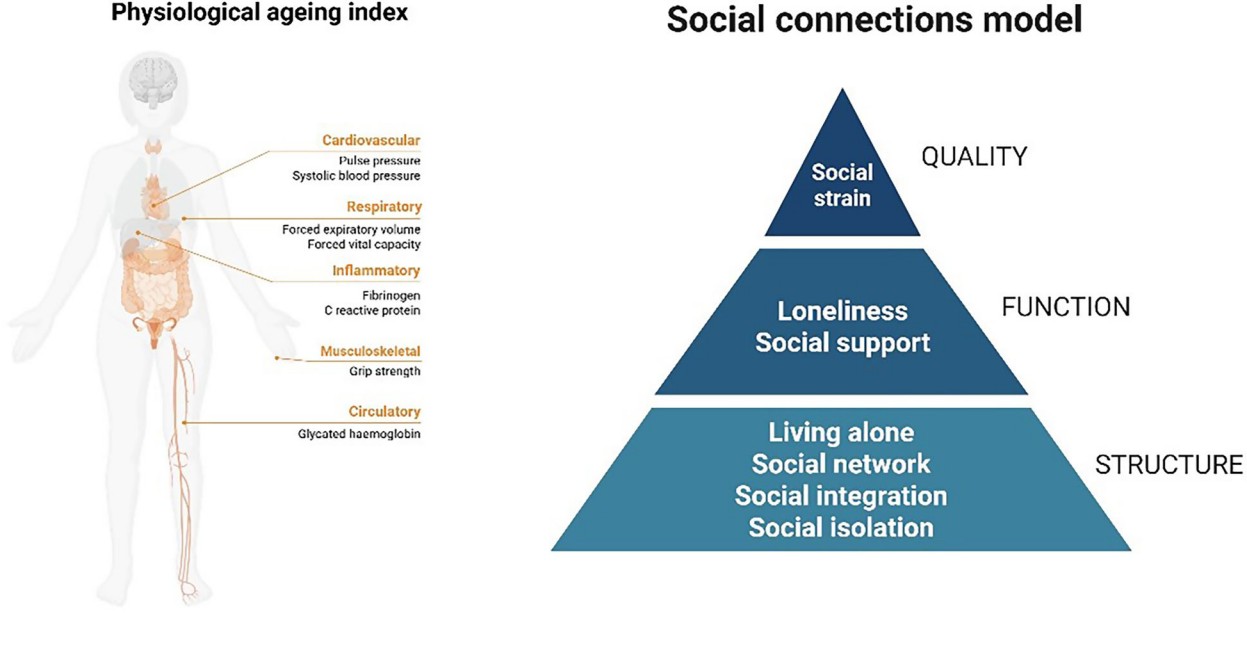

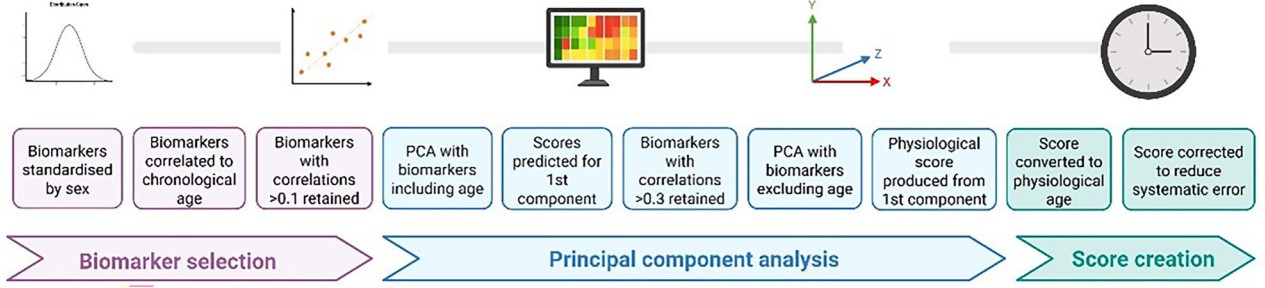

**Fig. 1 | Models of social connections and physiological ageing.** "Models of social connections and physiological ageing" created in BioRender. Fancourt, D. (2026) https://BioRender.com/qsgbn3k is licensed under CC BY 4.0.

**Quality**. High social strain did not have a statistically significant association with subjective age acceleration, but it did show some marginal associations with physiological age deceleration (1.2 years younger, 95%CI 0.2 to 2.1 years; although this did not survive correction for multiple comparisons).

### Sensitivity analyses
Overall, results were largely consistent across sensitivity analyses, with some results—such as small social network—more strongly associated with older physiological age when taking account of depression and outliers (Supplementary Figs. 3–5). Psychological age deceleration results were strongest for living alone in adults under the age of 65 and for social isolation for adults over the age of 65 (Supplementary Fig. 6). Notably, adults under the age of 65 did report feeling older if they had low levels of social integration. For physiological age acceleration, low social support was the clearest predictors in under 65 s, whereas all results from the main analyses were evident in over 65 s.

When considering whether strong social connections might be protective, living with others vs living alone showed the same findings as for social deficit analyses (just in reverse) given the binary nature of the variable (Fig. 4). No other factors were statistically significantly associated with subjective age deceleration, but high social integration and high social support but were both associated with physiological age deceleration. Those with high social integration were

physiologically 1.6 years younger (95% CI 0.5 to 2.7 years younger, 27% greater age deceleration than those without high social integration). When using subjective age and physiological age acceleration measures from four years later, living alone and low social support results for physiological ageing were maintained (Supplementary Fig. 7). When applying an alternative computation of proportional discrepancy in age, findings were materially unaffected (Supplementary Fig. 8). When using continuous variables for plausibly continuous confounders, the results were similarly materially unaffected (Supplementary Fig. 9).

## Discussion
Our findings show that our patterns of social connections are related in highly nuanced ways to subjective and physiological ageing. Weak structural aspects of social connections showed the strongest associations with objective measures of physiological age acceleration, with living alone and low social integration being risk factors for age acceleration. However, this was markedly at odds with people's perceptions of their ageing. No aspects of social connections were clearly related to older subjective age (apart from low social integration, which showed up in some sensitivity analyses but not others). But this is perhaps unsurprising given only 8% of people perceived themselves to be older than they were. Living alone was related somewhat counterintuitively to slower subjective age. Similarly, there was no

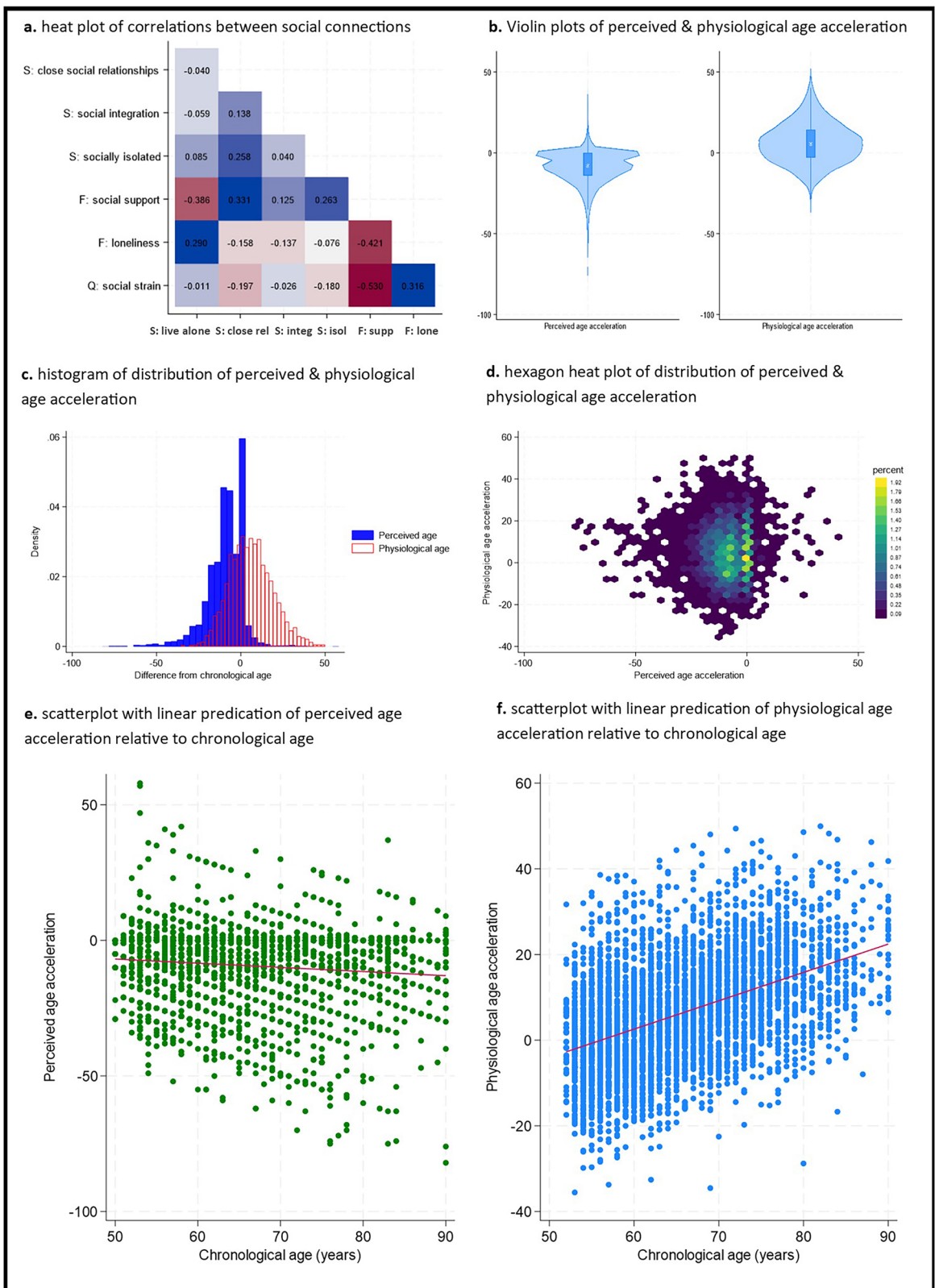

**Fig. 2 | Descriptive figures of exposure and outcome.** Correlations between exposure variables (**a**), distribution of exposure and outcome variables (**b–c**), the relationship between exposure and outcome variables (**d**), and outcome variables relative to chronological age (**e–f**).

statistically significant association between the functional and quality aspects of people's social connections and their subjective ageing, but low social support was associated with accelerated physiological ageing. These associations between weak social connections and physiological age acceleration were most prominent in adults as they became

older. In reverse, living with others and high social integration were associated with slower physiological ageing.

The relationship between weak social connections and accelerated physiological ageing echoes previous research showing associations between poor social connections and adverse physiological

**Table 1 | Sample descriptives**

| | Mean (SD)/Proportion (%) |
|---|---|
| Chronological age (years) | 65.6 (9.3) |
| Perceived age (years) | 56.1 (13.4) |
| Physiological age (years) | 71.4 (18.6) |
| **Sex** | |
| Male | 3155 (44.8%) |
| Female | 3892 (55.2%) |
| **Ethnicity** | |
| White | 6951 (98.6%) |
| Not White | 96 (1.4%) |
| **Working status** | |
| Not working | 4566 (64.8%) |
| Working full/part-time | 2481 (35.2%) |
| **Housing tenure** | |
| Other | 2574 (36.5%) |
| Outright homeowner | 4473 (63.5%) |
| **Physically inactive** | |
| No | 6648 (94.3%) |
| Yes | 399 (5.7%) |
| **Alcohol consumption** | |
| <5 times a week | 5162 (73.3%) |
| 5+ times a week | 1885 (26.7%) |
| **Current smoker** | |
| No | 6051 (85.9%) |
| Yes | 996 (14.1%) |
| **No. of chronic conditions** | |
| No | 5986 (84.9%) |
| Yes | 1061 (15.1%) |
| **Depression (CESD ≥ 3)** | |
| No | 5588 (79.3%) |
| Yes | 1459 (20.7%) |
| **Net non-pension wealth (quintiles)** | |
| 1 - lowest wealth quintile | 1305 (18.5%) |
| 2 | 1385 (19.7%) |
| 3 | 1430 (20.3%) |
| 4 | 1457 (20.7%) |
| 5 - highest wealth quintile | 1470 (20.9%) |
| **Educational attainment** | |
| Degree | 919 (13.0%) |
| NVQ3 A level/higher education | 2016 (28.6%) |
| NVQ2/GCE o level | 4112 (58.4%) |
| **No. of ADL limitations** | |
| 0 | 5795 (82.2%) |
| 1 | 698 (9.9%) |
| 2+ | 554 (7.9%) |
| **Self-reported health** | |
| Poor | 428 (6.1%) |
| Fair | 1316 (18.7%) |
| Good | 2269 (32.2%) |
| Very good | 2092 (29.7%) |
| Excellent | 942 (13.4%) |
| **BMI** | |
| BMI < 25 | 1949 (27.7%) |
| BMI 25–30 | 3054 (43.3%) |
| BMI ≥ 30 | 2044 (29.0%) |

outcomes, including for individual components of the physiological ageing index we used, such as cardiovascular risk factors, cognitive function, and inflammatory biomarkers[21–24]. But it extends it by showing that the associations exist with an aggregate index that has been independently and longitudinally demonstrated to have predictive potential for diverse patholoical age-related health outcomes. It is critical to note that the relationship presented here is likely bidirectional, with poor social connections leading to accelerated biological ageing, but also accelerated biological ageing in reverse reducing many types of social behaviours through reducing psychological or physiological capacity or reserves and making individuals more vulnerable to adverse health outcomes, as occurs in frailty[12,38]. However, It is important to note that our findings were independent of baseline chronic conditions and age-related disability and were preserved four years later. So we hypothesise that the relationship is not solely caused by a causal effect of accelerated physiological ageing attenuating social behaviours, and further hypothesise that accelerated physiological age may be a potential causal mechanism by which weak social connections could influence age-related morbidity and mortality. While this study (which focused only on cross-sectional and some initial longitudinal data) cannot make causal claims nor confirm such a mechanism, we put forward the hypothesis here as the basis for future research.

It is notable that structural factors had the strongest association with physiological ageing. In considerations of the relative risk of deficits to structural, functional and quality aspects of social connections, deficits to structure have been proposed as the most damaging, partly because structural factors underlie the capacity for an individual to experience good function or quality of relationships[21,39]. However, when considering which factors could be risk-reducing, only a subset of those factors whose deficits were adversely related to physiological age acceleration appeared protective when experienced in high quantities. Living with somebody was associated with the greatest age deceleration, but social integration was similarly protective. This could be because social integration involves diverse community and leisure activities that not only provide social connections but also bring additional salutogenic ingredients such as physical activity, cognitive stimulation, opportunities for multiple social identities, and often creativity and imagination, all of which are related to better health over time[40,41]. That said, it is important to note that these findings are likely bidirectional, as for many associations between social behaviours and health, with physiological functioning potentially not only influenced by weak social connections but also affecting one's ability to engage in social behaviours. Notably, when we used outcome measures from four years later, results maintained for living alone and low social support, but for social integration the effects were attenuated, likely because poor health can create barriers to social behaviours.

An important consideration is why there were such different results for subjective vs physiological ageing. Notably, there was almost no correlation between the two measures of age, with just 0.01% of the variance in one explained by the other. Only 8% of people perceived themselves to be older than their chronological age, compared to 67% being physiologically older on the index we used, which echoes previous work showing that after the age of 40, adults typically view themselves as younger than their age[42], and similar distinctions between subjective and biological age found in other studies[28]. In some instances, such as for living alone, there was a direct mismatch between subjective and physiological findings. It is possible that compensatory effects were at play here, with individuals who live alone having a lower personal need for social contact (e.g. due to personality type) or justifying these weak social connections internally through perceiving themselves to be better off alone. This compensatory effect could plausibly be reported for living alone but not other aspects of social connections as this is the most self-evident objective marker.

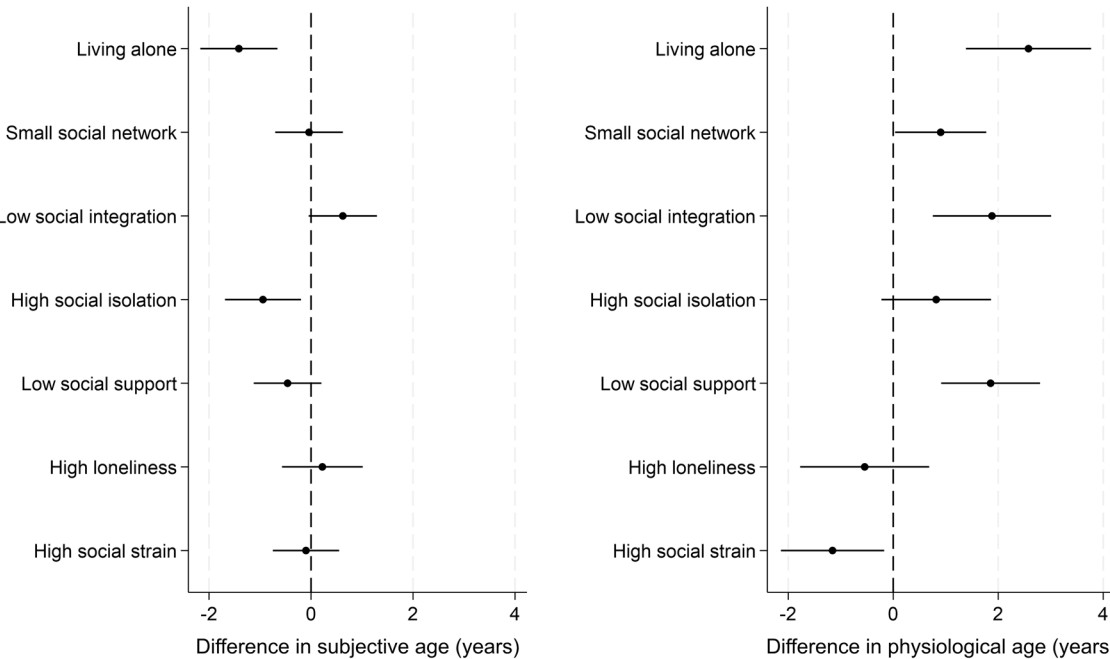

**Fig. 3 | Relationship between weak social connections and subjective age and physiological age acceleration (subjective ageing *n* = 6621, physiological ageing *n* = 4113; coefficient and 95% confidence intervals).** Average treatment effects from doubly robust estimations using the inverse-probability-weighted regression adjustment (IPWRA) estimator.

### Table 2 | Average treatment effects and 95% confidence intervals for main analyses

|  | Subjective age acceleration (ATE + 95% CI; years) | Physiological age acceleration (ATE + 95% CI; years) |
|---|---|---|
| Living alone | −1.41 [−2.17,−0.66] *p* < 0.001 | 2.58 [1.39, 3.77] *p* < 0.001 |
| Small social network | −0.04 [−0.70, 0.62] *p* = 0.908 | 0.90 [0.04, 1.77] *p* = 0.041 |
| Low social integration | 0.62 [−0.05, 1.29] *p* = 0.068 | 1.88 [0.75, 3.01] *p* = 0.001 |
| High social isolation | −0.94 [−1.69,−0.20] *p* = 0.013 | 0.82 [−0.23, 1.86] *p* = 0.124 |
| Low social support | −0.46 [−1.12, 0.20] *p* = 0.174 | 1.86 [0.91, 2.80] *p* < 0.001 |
| High loneliness | 0.22 [−0.57, 1.01] *p* = 0.582 | −0.54 [−1.77, 0.69] *p* = 0.386 |
| High social strain | −0.10 [−0.75, 0.55] *p* = 0.765 | −1.16 [−2.14, −0.17] *p* = 0.021 |

Average treatment effects from doubly robust estimations using the inverse-probability-weighted regression adjustment (IPWRA) estimator. *P* values are calculated from two-sided tests. A Bonferroni alpha of 0.006 was applied for multiple corrections, so results below this were maintained.
*ATE* average treatment effect, *CI* confidence interval.

It could also be that those who live alone have to stay more functionally independent to manage their own homes, and may have less experience of age-related discrimination from living with somebody younger, which is an experience related to subjective age[43,44]. Notably, previous studies have suggested that subjective age contributes to future emotional and physical status, but this relationship is not present the other way around[45]. Our results could suggest a similar effect at play for social status. However, such theories remain to be tested further through alternative research methods.

Overall, these findings may have a relevance for research focused on social connections and age-related pathology. Based on our findings that there is a stronger relationship with physiological than subjective ageing, we hypothesise that objective biological and functional processes may be more important mechanistically to the adverse health effects of weak social connections than broader processes related to subjective perceptions. The strength of the relationship with physiological ageing appeared to get stronger with chronological age, so we also hypothesise that the benefits of social connections (and interventions that increase these connections) may increase over time. Additionally, we also hypothesise that increasing physiological age may bidirectionally have a greater influence on social

behaviours as people become chronologically older. Future research is encouraged that tests these specific hypotheses in more detail, both through causal inference in longitudinal observational data and through experimental study.

Our study has many strengths including drawing its data from a large, representative sample of older adults, its theoretically-informed approach to exploring social connections, its use of a multi-dimensional model of physiological age that has been independently associated with diverse aspects of age-related pathology, its parallel consideration of subjective and objective processes in ageing, and its statistical use of doubly-robust estimators, which provide a clear advance on more traditional regression-based approaches to conditioning on counterfactuals. However, there are some limitations. As this was among the first studies into complex patterns of social connections and ageing, we focused on identifying a cross-sectional 'signature'. Nonetheless, we did undertake some explorations of the stability of this over the following four years, which suggested some of the strongest associations were maintained longitudinally. This provides a promising platform for further longitudinal work to establish if the associations presented here are causal. An important part of such future work will be able to explore whether temporal variation in confounders affects outcomes, whether effects are bidirectional (as we

hypothesise they likely are), and whether feedback effects are evident. In our analyses, we did not take account of survival. While for our cross-sectional analyses, this may have resulted in an under-estimation of effects, and while our longitudinal analyses showed commensurate results indicating limited effects of any potential survivor bias, future longitudinal analyses will require consideration of survival within models. Additionally, methods that account for unmeasured confounding are also recommended for future work to triangulate with findings with this study, where we acknowledge that even with two model specifications, residual confounding may remain. In additional limitations, PCA is a common method for biological age derivation, but it is by no means the only approach and has its own limitations, such as including chronological age within a first component, potentially capturing variance unrelated to ageing, and underestimating the hierarchical nature of biological processes. Further, although widely used, the aggregation of diverse physiological systems into a single ageing index is conceptually complex, with concerns that summary measures may obscure organ-specific heterogeneity in ageing trajectories. Nonetheless, we used this approach because it provides an interpretable summary metric that is straightforward to compare with chronological age and there is a strong precedent for its use[46–54]. We acknowledge other methods, including latent variables approaches or organ-specific ageing metrics, may better represent heterogeneity across systems so encourage future studies triangulating these findings with those from other approaches. Our sample also focused on community-dwelling adults, so it does not incorporate adults living in care homes, although this only accounts for around 2.5% of adults over the age of 65 in the UK[55]. We were limited by the measures of social connections included in the datasets. So we only had data on negative aspects of social quality, not positive ones. We specifically chose to focus on the bottom quartile of respondents as there are no official cut-offs recommended for many of the scales, so this provided a consistent approach across all measures in the study. Dichotomous variables do bring challenges, including loss of power and concealment of non-linearity within relationships[56]. and potential for remaining confounding. And we acknowledge that the lack of statistically significant association between lowest quartiles and our outcomes could also reflect differences in the distribution of participants across the possible spectrum of experiences. However, we conducted analyses that used continuous variables for confounders to minimise the challenges of categorical variables and applied dichotomous variables for our exposure strategically, in line with recommendations from social epidemiology to define exposures in a way that provides well-defined hypothetical interventions[57]. We still recommend future studies that look at social connections as continuous exposures, but from a public health perspective, our analyses also suggest a value in exploring more specifically the role of interventions targeted at those experiencing weakest social connections within society. Finally, In this study, we focused on an overall index of physiological age, in keeping with other ageing research[46–54], but future studies are encouraged that look at organ-specific ageing to ascertain whether social deficits affect different biological systems at different rates.

In conclusion, we found no credible evidence that having friends helps older adults to stay younger either in terms of their subjective or physiological age. But lack of support from friends and other family and relatives and weaker integration into social and community activities is associated with faster physiological ageing, even if we perceive the contrary ourselves. These findings raise important hypotheses about biological mechanisms that could help explain the relationship between social deficits and age-related morbidity and mortality outcomes. Future studies are encouraged to test whether building the structural foundations for good social connections may be a risk-reducing strategy for physiological age acceleration. Given the discrepancy between people's subjective and physiological ageing, individuals may be unaware of the underlying deficits and benefits of social connections, so

enhancing awareness of the impact that social connections have on our health remains an important public health strategy.

## Methods
### Dataset
Data were derived from the English Longitudinal Study of Ageing (ELSA). ELSA is a nationally-representative panel study of people aged 50 and over and their partners, living in private households in England. The original sample was drawn from participants from the Health Survey in England (HSE) in 1998, 1999 and 2001[58]. Survey administration is done via computer-assisted personal interviewing. This research complies with all relevant ethical regulations. ELSA received ethics approval most recently from the South Central – Berkshire Research Ethics Committee (23/SC/0112). Written informed consent was obtained at each interview. No further ethics approval was required for the present study. The first wave of data collection commenced in 2002/2003, and participants have been followed biennially since. We used wave 2 (2004/5), as this wave included a rich battery of social and physiological variables of relevance.

We restricted participants to core ELSA members who had returned the self-completion questionnaire where most of our main variables of interest were measured ($n = 8354$), who were over the age of 50 ($n = 8129$), and who had full data on exposures and confounders ($n = 7047$). Within this sample, 6621 had data on subjective ageing, and 4113 had full data on physiological ageing. In main analyses, the final sample was allowed to vary between analyses for subjective and physiological age to maximise power, but sensitivity analyses restricted the sample just to individuals who had both subjective and physiological ageing ($n = 3878$).

### Exposures
**Structural.** Structural aspects of social connections included living alone (vs with others), intimate network size (how many children, family and friends respondents had close relationships with; count variable split into lowest quartile indicating most deficit vs other quartiles), social integration (number of organisations, clubs and societies respondents were a member of, their engagement with diverse volunteering activities, and their engagement with diverse cultural activities; count variable split into lowest quartile of integration vs other quartiles), and social isolation (whether respondents saw or spoke on the phone to family, friends or children less than once a week, resulting in a scale for each from 0 to 6, with the scores averaged and split into most isolated quartile vs other quartiles)[59].

**Functional.** Functional aspects of social connections included loneliness (measured using the three-item subscale from the revised UCLA loneliness scale, resulting in a scale from 3 to 9 split into most lonely quartile vs other quartiles [22]), and subjective social support (whether participants feel they can rely on others, can open up to them, and are not let down by them, asked separately for partner, children, relatives and friends with each question scored from 1 to 4, with the scores averaged and split into lowest social support quartile vs other quartiles).

**Quality.** Quality aspects of social connections included social strain (whether participants feel they are criticised by others, not understood by them, and they get on their nerves, asked separately for partner, children, relatives and friends with each question scored from 1 to 4 generating a scale of 12–48 split into highest strain quartile vs other quartiles)[59].

Social connections measures are summarised in Fig. 4 and full details of questions are provided in Supplementary Table 1. Figure 4 adapts the pyramidal structure previously proposed in theoretical work on social connections that positions structural factors as underpinning and being critical to the experience of functional or quality aspects of social connections[39].

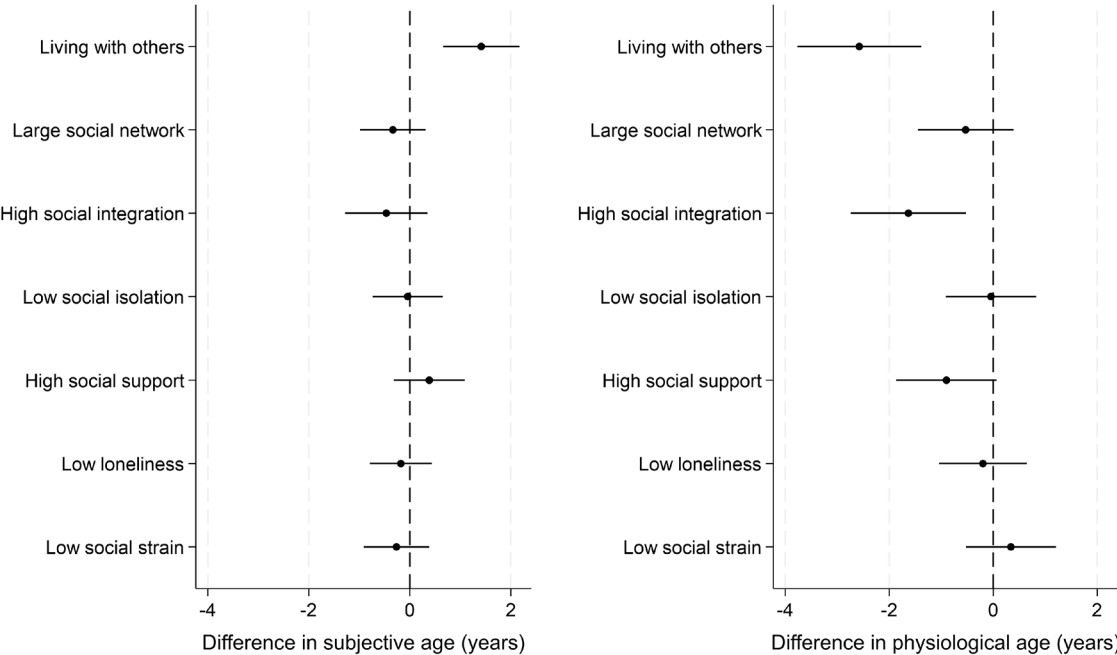

**Fig. 4 | Relationship between positive social connections and subjective age and physiological age acceleration (subjective ageing *n* = 6621, physiological ageing *n* = 4113; coefficient and 95% confidence intervals).** Average treatment effects from doubly robust estimations using the inverse-probability-weighted regression adjustment (IPWRA) estimator.

## Outcomes

Subjective age was derived by asking participants how old they felt (measured in years) and subtracting chronological age from this. As a result, negative scores indicate older subjective age, and positive scores indicate older subjective age.

A physiological age acceleration measure (measured in years) was derived using the principal component analyses (PCA) method, an established method to measure physiological ageing[46,60]. It was created using clinical indicators pertaining to the cardiovascular system (pulse pressure, systolic blood pressure, diastolic blood pressure, mean arterial pressure), respiratory system (forced vital capacity [FVC], forced expiratory volume in one second [FEV]), the haematologic system (haemoglobin concentration, fibrinogen, C-reactive protein [CRP], ferritin), metabolic system (fasting glucose, glycated haemoglobin, total cholesterol, LDL cholesterol, HDL cholesterol, triglyceride), and muscle and fat (grip strength, waist circumference). All measures were taken at wave 2 in nurse visits (blood-based biomarkers and physiological testing). This index followed the process previously described for creating a physiological ageing index[27].

The full details of physiological age derivation using the PCA method are summarised in Fig. 4 and Supplementary Materials (Appendix 1; Supplementary Table 2 and Figs. 10 and 11). To create the physiological age acceleration/deceleration measure, chronological age was subtracted from physiological age; an approach extensively used in previous research[47,61–63]. As in previous studies, this means that accelerated physiological ageing refers to a chronological age higher than a physiological age rather than necessarily imply a longitudinal change. Negative scores indicate decelerated ageing (slower physiological ageing), and positive scores indicate accelerated ageing (faster physiological ageing). Analyses of this index show its association with incident functional and mobility limitations, memory impairment, and diverse chronic conditions in the ELSA study population (Supplementary Table 3).

## Confounders

Factors identified as predicting both receptive arts engagement and mortality were identified using directed acyclic graphs (DAGs) and included as covariates[64]. Demographic and socio-economic confounders included chronological age (in years), sex (male or female; ELSA survey materials refer to sex not gender), ethnicity (white British vs other), educational qualifications (no educational qualifications; education to GCE/O-levels/national vocational qualification (NVQ) 2 (qualifications at age 16); education to NVQ3/GCE/A-levels (qualifications at age 18); higher qualification/NVQ4/NVQ5/degree), total non-pension wealth (which combines net financial and physical wealth plus net owner-occupied housing wealth; categorised in quintiles)[65], and house ownership (whether individuals owned their property outright vs with a mortgage/renting/social housing/other).

Health and behavioural factors were more complex to theorise within models as they can be both predictors of social connections and mediators of causal relationships to ageing-related outcomes. However, longitudinal research and the results of clinical trials generally show that social relationships are relatively weak predictors of future health behaviours, while health behaviours more strongly predict social connections[66–68]. So we included whether participants were sedentary (categorised as engaging in mild, moderate or vigorous activity less than once a week), frequency of alcohol consumption (more than 5 days a week vs less), and whether participants currently smoked. We also acknowledge a bidirectional relationship between social connections and disease[21]. However, given that chronic illnesses have marked effects on physical and psychological capacity to undertake social behaviours, we included whether participants reported currently having a diagnosis of any chronic conditions including cancer, lung disease or cardiovascular disease (including high blood pressure, angina, a previous heart attack, heart failure, a heart murmur, an abnormal rhythm, diabetes, a previous stroke, high cholesterol, or other heart trouble), arthritis, asthma, osteoporosis, Parkinson's disease, Alzheimer's disease or dementia. But this variable was provided as a simple binary to acknowledge the presence of a capacity-influencing chronic illness while minimising the risk of accounting for partial mediation of the effect of the exposure on the outcome.

More complex potential confounders were difficulties in carrying out activities of daily living that participants reported (ADLs; count of

challenges including dressing, bathing, eating, using a toilet, shopping, taking medications or making telephone calls), self-reported health (poor, fair, good, very good, excellent), BMI (<25, 25–30, ≥30), and depression (using the Centre for Epidemiological Studies Depression (CES-D) scale, ≥3). For all of these, literature suggest strong bidirectional relationships, particularly for BMI and depression in relation to social connections, with certain aspects of social connections such as loneliness sharing highly overlapping genetic influences[17,69]. Additionally, ADLs and self-reported health could be argued to overlap with components of our outcome measures. Consequently, we conducted models with and without these factors to test the consistency of findings, as outlined below.

### Statistics

Data were analysed using doubly robust estimation using the inverse-probability-weighted regression adjustment (IPWRA) estimator. This involves building two models to account for the non-random treatment assignment: a regression adjustment model for the outcome and a treatment-assignment model for the exposure, only one of which has to be correctly specified, enhancing the robustness of the analysis[70]. This approach thus provides two chances to make valid inferences, instead of just one, as long as either the treatment group or outcome can be determined as a function of the covariates and there are no unobserved confounders. IPWRA estimators apply weighted regression coefficients to compute averages of treatment-level predicted outcomes, where the weights are the estimated inverse probabilities of treatment. Following our DAGs, confounders applied to both exposure and outcome were sex, ethnicity, wealth, education, house ownership, sedentary behaviours, alcohol consumption, smoking, and chronic conditions. Problems with ADLs and self-reported health were additionally applied to the exposure, and BMI and depression to the outcome.

We estimated the average treatment effect in the population (ATE), which is the average difference in outcome if everyone in the population experienced the exposure, versus no-one in the population. We also estimated the potential-outcome means (POMs) by regression adjustment, which use contrasts of averages of treatment-specific predicted outcomes to estimate treatment effects, and used this to calculate the ATE as a percentage of the untreated POM (presented with the delta-method-based standard error)[71]. Separate models were estimated for each of the eight exposures, so a Bonferroni alpha of 0.05/8 or 0.006 accounts for multiple corrections. For all analyses, we applied probability weights for the self-completion questionnaire provided in the data, which account for complex sampling strategies and non-response. All analyses met model assumptions and all tests conducted were two-tailed. Analyses were carried out in Stata v18.

Depression could plausibly lie on the causal pathway from social connections to ageing, so it was only included in the outcome confounder model, not the exposure confounder model, in the main analyses. However, a sensitivity analysis additionally accounted for depression within the exposure model. A second sensitivity analysis restricted the sample to only people who had data both for the subjective and physiological ageing indices. A third sensitivity analysis excluded anybody whose subjective or physiological age was more than 30 years above or below their chronological age to assess the stability of the findings without the influence of outliers. A fourth sensitivity analyses dichotomised the results by age (split at 65 years) in order to ascertain whether results were consistent across ageing. A fifth sensitivity analysis recategorized the quartiles such that the quartile showing least sign of social deficit and most sign of strong social connections was the exposed population to ascertain whether the findings were the same for strong rather than weak social connections. A sixth sensitivity analysis repeated the main analyses but using subjective age and physiological age acceleration measured four

years later, to assess whether findings held longitudinally as well as cross-sectionally. A seventh sensitivity analysis repeated the main analysis using an alternative algorithm to calculate proportional discrepancy in age: age index-chronological age/chronological age[44]. An eighth sensitivity analysis also repeated the main analysis but removing cognitive measures form the physiological ageing index to provide standardisation with previous papers. A final sensitivity analysis considered the consistency of results when using continuous variables for plausibly continuous confounders.

### Reporting summary

Further information on research design is available in the Nature Portfolio Reporting Summary linked to this article.

## Data availability

ELSA data are available from the UK Data Service https://doi.org/10.5255/UKDA-SN-5050-35.

## Code availability

Code is available from https://doi.org/10.5281/zenodo.17986826.

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

## Acknowledgements

This research was funded by the UK Research and Innovation [MR/Y01068X/1] and grants for the English Longitudinal Study of Ageing (NIH: R01AG017644 and NIHR: 198-1074).

## Author contributions

M.B. and A.S. derived the physiological ageing index. D.F. and A.S. designed the current study. D.F. ran the analyses and drafted the manuscript. All authors critically appraised the manuscript and approved it for submission.

## Competing interests

The authors declare no competing interests.
