## [Transparent Peer Review file · Nature Communications]

Social connections are differentially related to perceived and physiological age acceleration amongst older adults

Corresponding Author: Professor Daisy Fancourt

Version 0:

Reviewer comments:

Reviewer #1

(Remarks to the Author)

The manuscript is well-written, with a strong and rigorous methodology that enhances the credibility of its findings. The authors have effectively utilized robust analytical approaches to examine the association between social connections and physiological aging. The study makes a valuable contribution to the field by highlighting the role of structural aspects of social connections in shaping biological aging processes. I have a few suggestions for a minor revision to further strengthen the manuscript.

Methods:

- There are no references provided for structural aspects of social connection, as well as social support and strain. Please include relevant citations to support these constructs.
- Please modify “gender (male and female)” to “sex” for accuracy in terminology.

Discussion:

This study provides novel insights by demonstrating that structural aspects of social connections—such as living alone or lacking social integration—are most strongly associated with physiological aging. However, frailty—a key age-related state characterized by increased vulnerability to stressors—may also play a crucial role in this mechanism, yet it was not explicitly considered in the conceptual framework. Given that both social isolation and frailty are stressful conditions that activate inflammatory pathways and contribute to health deterioration, frailty could serve as an important explanatory factor in the observed associations. Since frailty is closely linked to systemic inflammation, immune dysregulation, and physiological decline, it may help explain how social deficits accelerate biological aging.

While the authors have controlled for multiple indicators of frailty (e.g., ADL, IADL, chronic conditions, and cognitive function), explicitly framing frailty as a key part of the theoretical model could enhance the interpretation of the findings. Given its bidirectional relationship with social isolation—where socially isolated individuals are at higher risk of developing frailty, while frailty itself can lead to further social withdrawal—it may serve as a key mechanism linking social deficits to accelerated biological aging. Addressing frailty within the discussion would strengthen the study’s contribution to understanding the complex interplay between social connections and physiological aging.

(Remarks on code availability)

The supplementary files are very comprehensive; however, I did not see any code included. If the journal’s policy requires sharing code, it may be helpful to provide the code used for the Principal Component Analysis (PCA) to enhance transparency and reproducibility.

Reviewer #2

(Remarks to the Author)

The paper examines the relationship between social deficits and ageing, using perceived and physiological age as outcome measures. While the study benefits from a large dataset and the inclusion of multiple biological and self-reported indicators, there are substantial conceptual and methodological weaknesses that limit the validity of its conclusions. Key concerns include ambiguous causal claims, inconsistent variable classification, methodological weaknesses in subjective age

calculation; questionable categorization of continuous variables; unclear mechanistic claims. Given these issues, the interpretability and robustness of the findings remain questionable.

- Abstract & throughout the ms: The manuscript states that the study provides insights into ageing-related mechanistic processes, yet the analysis primarily examines associations between perceived age, physiological age, and social support. There is no clear justification for how these associations inform underlying mechanisms of ageing. Correlational findings do not establish causal pathways or explain biological or psychological processes that drive ageing. Without a theoretical or empirical framework linking these variables to mechanistic ageing processes, the claim remains unsupported. The authors should either (1) clearly articulate how their findings contribute to mechanistic understanding or (2) revise their conclusions to reflect that the study identifies associations rather than mechanisms.
- Abstract & throughout the ms: The term "social deficits" to describe lack of social support, loneliness, and related constructs is problematic, as it implies a deficiency-based or pathological framing of social experiences. These are context-dependent and multidimensional and should not be reduced to a deficit model without justification. The authors should use more precise terms, such as "social support," "social isolation," or "social networks," to align with established terminology.
- The manuscript contains several typographical errors, such as "metaboloic" (Abstract). A thorough proofreading is necessary to ensure accuracy and clarity throughout the text.

INTRODUCTION

- The claim that "social deficits increase with age" oversimplifies social changes in later life. Research, including Socioemotional Selectivity Theory (SST), shows that older adults prioritize quality over quantity in relationships rather than experiencing a uniform decline. Reduced social contact does not necessarily indicate a deficit. The authors should consult existing theories on social support across the lifespan and revise this statement to reflect these nuanced patterns.
- The introduction is lengthy in parts, particularly in its deep dive into the history of biological ageing, yet it lacks a clear trajectory toward the study's main research question as there is no link to established physiological. As a result, the reader is left uncertain about the study's focus and contribution. The authors should streamline the intro, ensuring that each section directly builds toward the study's aims.
- Again, one of the biggest issues is the misuse of causal language in an observational study, such as stating that "biological age has been strongly influenced by social deficits." This implies a directional effect that has not been empirically established. The authors should use association-based language (e.g., "is correlated with") unless they provide strong theoretical and methodological evidence to support causal claims. Ensuring precise language is essential for clarity and validity.
- The term "perceived age" is too ambiguous, as it is often used to describe how others perceive a person's age, rather than an individual's own subjective experience of ageing. If the authors are referring to how old participants feel rather than how old they appear to others, the more precise term would be "subjective age" or "felt age". Clarifying this terminology would prevent misinterpretation and align with established literature on self-perceptions of ageing.
- Referring to perceived age as a biomarker is conceptually problematic. While perceived age can be an important predictor of health outcomes and mortality, it does not meet the standard definition of a biomarker, which refers to objective, biological measures. Perceived age is a subjective or socially influenced measure rather than a direct biological marker of ageing. The authors should reconsider this classification.
- The claim that "perceived age provides a mental map of one's remaining life expectancy" is overstated. While subjective age correlates with subjective life expectancy, they are distinct constructs and do not perfectly overlap. The authors should revise this statement to reflect their partial but imperfect association.

METHODOLOGY

- A major methodological issue in the paper is the categorization of continuous variables into extreme groups (e.g., lowest vs. quartiles). Dichotomizing or grouping continuous measures, such as depression (yes, no); activity (yes no), etc. , leads to a significant loss of information, reduces statistical power, and may introduce artificial thresholds that do not reflect the underlying distribution of the data. If the variable is inherently continuous, the authors should avoid arbitrary splits and instead model it as a continuous predictor to retain the full variability and improve analytical precision.
- The term "perceived age acceleration" is misleading and conceptually incorrect in this context. The data only capture a single time point, meaning it is impossible to determine whether participants have experienced a change/acceleration in the discrepancy between subjective age and chronological age over time. Acceleration implies an increasing discrepancy over multiple time points, which is how the term is used in the existing literature. If the difference between subjective and chronological age is stable, it does not reflect acceleration. It's possible that some people have always felt this way and this is a trait with zero acceleration. The authors should either revise the terminology to avoid implying change over time or incorporate longitudinal data if acceleration is the intended construct.
- Same is true for the term "physiological age acceleration". It is misleading and conceptually incorrect as the data only capture a single time point, meaning it is impossible to determine whether participants have experienced a change/acceleration in the discrepancy between physiological age and chronological age.

- The calculation of subjective/felt age as "perceived age minus chronological age" does not appropriately account for age-related scaling differences. A one-year discrepancy is more meaningful for a 20-year-old than for a 90-year-old. To address this, most studies compute a proportional discrepancy by dividing the subjective age difference by chronological age (i.e., $(\text{Subjective Age} - \text{Chronological Age}) / \text{Chronological Age}$), which better accounts for relative differences across the lifespan. The authors should consider aligning their approach with this established method (Rubin and Berntsen 2006; Stephan et al. 2015) to improve comparability with existing literature. Also, proportional discrepancy scores three standard deviations above or below the mean need to be considered outliers (Stephan et al. 2015; Weiss and Lang 2012) and replaced.

- The approach used to derive the physiological age acceleration measure raises several methodological concerns:

- o Heterogeneous Temporal Dynamics: The included physiological systems (cardiovascular, metabolic, cognitive – is this even physiological?), etc.) exhibit different rates of change across the lifespan. Some measures (e.g., blood pressure, grip strength) might show a more linear age-related decline, while others (e.g., cognitive function) might exhibit nonlinear trajectories and are influenced by cumulative exposures. Combining these without considering their distinct temporal orders may lead to severe misinterpretation of ageing processes.

- o Causality and Interdependencies: The included indicators are not independent—some serve as causes, others as consequences of physiological ageing. For example, metabolic dysfunction may contribute to cardiovascular deterioration, which in turn affects cognition. Ignoring these interdependencies may distort the meaning of the derived physiological age acceleration scores.

- o Principal Component Analysis (PCA) seems inappropriate → PCA assumes that the included variables represent independent dimensions of variation, but ageing is a highly correlated, hierarchical process. Using PCA for dimensionality reduction may mix cause and consequence, and the first component may capture variance unrelated to ageing per se. Alternative methods such as latent variable modeling (e.g., structural equation modeling) could better account for hierarchical dependencies.

- o The authors include chronological age in the PCA, predict component scores, examine correlations, and then remove chronological age in a second PCA to compute physiological age. This approach introduces two key issues. First, including chronological age in the first PCA forces the first component to capture chronological age variance, making it unclear whether the component represents ageing-related physiological changes or just chronological age itself. Second, By running PCA again without chronological age, the method disconnects the derived score from the variable initially used to define biomarker importance. This creates an inconsistency in how ageing is defined and how the final score is constructed.

- o The calculation of physiological age acceleration (physiological age – chronological age) assumes a linear relationship between physiological and chronological ageing. However, this assumption is problematic given the nonlinear nature of biological ageing. Some physiological systems may decline early (e.g., lung function), while others remain stable longer (e.g., cognitive function), making a single age difference score potentially misleading.

- o While the authors state that the index is associated with functional limitations and chronic conditions, they should provide rigorous validation analyses, including: Comparison with existing biological age models (e.g., epigenetic clocks, telomere length, brain age, skin age, etc. etc.).

- o Biomarkers were pre-selected based on their correlation with chronological age ($r \geq 0.10$), ensuring that the final physiological age score is artificially correlated with age. This approach excludes biomarkers that may reflect physiological ageing independent of chronological age, potentially skewing results. It is possible that an indicator is not highly correlated at one point in the lifespan but yet predictive for mortality outcomes.

- o The chosen correlation cutoffs ($r \geq 0.10$ for selection, etc.) appear arbitrary and lack justification. A weak correlation with age ($r \geq 0.10$) does not ensure that a biomarker reliably captures physiological ageing. A moderate correlation ($r \geq 0.30$) with the PCA score does not confirm biological relevance, as PCA prioritizes variance, not ageing-specific processes.

- o Most importantly, PCA assumes that all retained physiological measures the same underlying construct, ignoring the heterogeneous and multi-system nature of ageing. Ageing affects different systems (e.g., metabolic, immune, cardiovascular) at different rates; reducing it to a single component oversimplifies complex biological processes and ignoring the heterogeneous nature of ageing across different organ systems.

- The use of doubly robust estimation with the IPWRA estimator in a cross-sectional study raises some methodological concerns. First, the lack of temporal ordering prevents causal inference, as exposure and outcome are measured simultaneously. Second, unobserved confounders may bias the results. Third, inverse probability weights (IPWs) are more appropriate for longitudinal studies, where they adjust for selection bias over time; their use in a cross-sectional setting is less effective. Fourth, many covariates, including sex and health behaviors, vary over time, but the analysis relies on a single time-point measurement. Finally, the cross-sectional design limits the ability to conduct sensitivity analyses to assess time-dependent confounding or feedback effects.

- The classification of variables as confounders, mediators, or outcomes appears arbitrary and lacks clear conceptual and empirical justification. For example, depression could plausibly act as a mediator, confounder, or bidirectional factor in the relationship between social connections and ageing, depending on the theoretical framework. The decision to include certain variables in the exposure vs. outcome confounder models seems inconsistent and requires a stronger theoretical rationale.

- Table 1 lists "No. of chronic conditions" as "Yes/No," which seems wrong—a count variable cannot be binary. The authors should clarify whether they measured the number of conditions or the presence of any condition and adjust the categorization accordingly.

- The manuscript lacks a full intercorrelation table for all study variables, except for social support. Given the potential for multicollinearity, particularly among ageing-related psychological measures, the authors should provide a full correlation matrix to assess the extent of redundancy and ensure the validity of their models.
- The pyramid structure used for the social connection model implies an implicit hierarchy or ordering of social relationships that lacks empirical justification. If the model suggests qualitative differences between levels, the authors should provide empirical support for this structure. Otherwise, an alternative visualization that does not imply a strict ranking may be more appropriate.
- Resolution of figures is poor.

DISCUSSION:

- Additionally, the literature review is highly selective, overlooking key studies in ageing, social support, and biological ageing. The authors should engage more comprehensively with existing research to provide a balanced and accurate foundation for their work.
- The claim that the study provides "novel insight into a plausible mechanism" linking social deficits to ageing is overstated and inaccurate. Demonstrating an association between social support and a composite ageing index does not establish a mechanism, as mechanisms require causal pathways explaining how social factors biologically influence ageing. The authors should clarify their contribution—if they are identifying correlations, they should not claim mechanistic insight. Additionally, the novelty of this finding is questionable, given prior research on social determinants of (subjective and biological) ageing.
- Similarly, the claim that these findings "provide insight into how social deficits influence age-related pathology" is overstated. Additionally, the statement that "objective biological and functional processes may be more important mechanistically than broader processes related to subjective perceptions" is unclear and lacks empirical justification. The authors should clarify their reasoning, define what is meant by "broader processes," and avoid mechanistic claims.

(Remarks on code availability)

Did not see the code.

Reviewer #3

(Remarks to the Author)

Authors used ELSA data to evaluate social connections (structural, functional, and quality) on perceived and physiological aging. Overall, the study was well written and contributes to the literature in a meaningful way. However, I have a few recommendations to strengthen the contribution and justification for the design of the study.

Introduction:

I would recommend rewording the lines 44-54 as it makes it seem that physiological age is new – even though it precedes the development of epigenetic clocks and metabolic clocks. I would perhaps also focus on the fact these types of clocks were not trained on prediction of outcomes but rather incorporate physiological aging processes themselves (unlike epigenetic clocks that were trained on outcomes).

I am uncertain about the current motivation for perceived age and physiological age. It seems like two different papers to me. I would recommend some additional incorporation that may answer is one related to health better than the other? How are they related to each other in other studies since it has low correlation here? Why are both necessary to understand the mechanistic processes? Giving further motivation here would help to strengthen the contribution of the paper and highlight why we need to consider perceived (even if you the authors do not find as robust findings).

Methods:

Can the authors motivate the addition of cognitive functioning as a component of physiological age, as cognition is not purely a physiological process but a combination of cognitive development and pathology. To justify this use, I might also recommend the authors do a sensitive check without cognitive functioning to keep in line with prior iterations. I do recognize that it is unlikely to change the results as cognitive functioning often runs parallel to the other components, but conceptual clarity would be helpful here.

Discussion:

What does the role of mortality selection have on these models? Is this a limitation to consider? I appreciate the 4 years follow up but I am still wondering about sample selection and survival and important processes to consider.

Could the authors also add that people with health conditions may also have more complications with social participation, which itself may lead to faster aging? I realize that the living alone and living with someone may partly address this concern (especially with severe health conditions) but feel it may be an important note to consider for future work (or even a potential sensitivity check).

Minor comments:

You write the first aging clocks in the introduction. I would reword to state the first-generation epigenetic aging clocks were trained on chronological age. While not called aging clocks per se, other algorithms like PhenoAge served a similar purpose and may confuse readers.

Please specify if sample was community dwelling adults. If so, that may also be a limitation to consider. But I am uncertain of the commonality of nursing homes in the UK context.

(Remarks on code availability)

Version 1:

Reviewer comments:

Reviewer #1

(Remarks to the Author)

The authors have addressed my comments!

(Remarks on code availability)

The authors provided comprehensive codes, including the creation of scales, execution of the main models, and sensitivity analyses. Excellent work!

Reviewer #2

(Remarks to the Author)

I appreciate the authors responsiveness. The authors acknowledge the concerns raised. However, the manuscript has the potential for deeper improvement.

Page 2; first paragraph: The revised framing improves clarity but still misrepresents the state of the literature. The claim that physiological aging indices are “providing new avenues” overstates their novelty. These indices have been developed and used for over a decade, and many researchers explicitly avoid combining heterogeneous systems (e.g., cardiovascular, inflammatory, functional) into a single composite due to their distinct biological trajectories and limited conceptual coherence. Moreover, the field has moved toward more mechanistically grounded approaches such as epigenetic clocks, which may reflect upstream aging processes more precisely. The revised paragraph would benefit from a more balanced acknowledgment that physiological indices are not conceptually new, and their aggregation remains contested. This would also temper the overstated novelty in linking them to social variables—something that has been explored, if not always in aggregated form.

Page 2; second paragraph: The claim that subjective and physiological age provide “complementary insight into mechanistic processes” remains rather speculative without theoretical or empirical justification. A more cautious and theoretically grounded framing is needed to clarify the study’s actual contribution.

The authors acknowledge concerns about heterogeneous aging dynamics but largely dismiss them by relying on visual inspection using local polynomial regression. While the added plots in the Supplementary Materials are appreciated, the justification remains insufficient. Local polynomial smoothing is, as the authors note, prone to overfitting and does not provide formal tests of nonlinearity. More importantly, visual linearity within a constrained age range does not negate the conceptual issue: the biomarkers reflect distinct physiological systems with inherently different temporal trajectories and exposures. Simply showing approximate linearity does not resolve the concern about combining them into a single metric.

The inclusion of memory as part of the physiological aging index remains conceptually problematic. While cognitive decline is age-related, it is not equivalent to brain age, nor is it typically considered a direct marker of physiological aging. Brain age clocks rely on structural and functional neuroimaging or molecular data—distinct from behavioral measures of memory performance. Equating memory scores with brain age in their line of reasoning conflates separate constructs. Although the authors conducted sensitivity analyses excluding cognition, the rationale for its inclusion still lacks theoretical clarity. The manuscript should more explicitly acknowledge these conceptual distinctions and avoid overstating the relevance of memory as a physiological indicator.

Relatedly, the term “memory” is too general to accurately describe the cognitive measures used in the index. Tasks assess specific aspects of episodic memory. Given the differences between memory domains—and their distinct aging trajectories—the manuscript should specify that the measure reflects episodic memory performance.

Figures 3 & 4: The x-axis label needs to be updated to the new terminology - “differences in perceived age (years)”

Page 12, 2nd paragraph: The revised section introduces useful nuance by acknowledging bidirectionality and the limits of causal inference, but it still leans too heavily on speculative interpretation. Moreover, positing this as a “potential causal mechanism” risks overstating the implications of what remains correlational evidence—particularly given the composite nature of the physiological index and the lack of temporal resolution. I recommend more cautious wording that emphasizes hypothesis generation rather than mechanistic interpretation.

(Remarks on code availability)

Reviewer #4

(Remarks to the Author)

-

(Remarks on code availability)

-

Response to reviewers

Reviewer #1 (Remarks to the Author):

The manuscript is well-written, with a strong and rigorous methodology that enhances the credibility of its findings. The authors have effectively utilized robust analytical approaches to examine the association between social connections and physiological aging. The study makes a valuable contribution to the field by highlighting the role of structural aspects of social connections in shaping biological aging processes. I have a few suggestions for a minor revision to further strengthen the manuscript.

We are grateful for the positive feedback from the reviewer.

Methods:

- There are no references provided for structural aspects of social connection, as well as social support and strain. Please include relevant citations to support these constructs.

We have now provided more citations for these measures.

- Please modify “gender (male and female)” to “sex” for accuracy in terminology.

This has now been done

Discussion:

This study provides novel insights by demonstrating that structural aspects of social connections—such as living alone or lacking social integration—are most strongly associated with physiological aging. However, frailty—a key age-related state characterized by increased vulnerability to stressors—may also play a crucial role in this mechanism, yet it was not explicitly considered in the conceptual framework. Given that both social isolation and frailty are stressful conditions that activate inflammatory pathways and contribute to health deterioration, frailty could serve as an important explanatory factor in the observed associations. Since frailty is closely linked to systemic inflammation, immune dysregulation, and physiological decline, it may help explain how social deficits accelerate biological aging. While the authors have controlled for multiple indicators of frailty (e.g., ADL, IADL, chronic conditions, and cognitive function), explicitly framing frailty as a key part of the theoretical model could enhance the interpretation of the findings. Given its bidirectional relationship with social isolation—where socially isolated individuals are at higher risk of developing frailty, while frailty itself can lead to further social withdrawal—it may serve as a key mechanism linking social deficits to accelerated biological aging. Addressing frailty within the discussion would strengthen the study’s contribution to understanding the complex interplay between social connections and physiological aging.

Thank you for this point. In response to comments from the other reviewers, we have now added a new paragraph on the directionality of our effects in the discussion and we explicitly mention frailty as a potential explanatory factor through which accumulation of biological ageing and loss of reserves leads to changes in social behaviours (discussion paragraph 2).

Reviewer #1 (Remarks on code availability):

The supplementary files are very comprehensive; however, I did not see any code included. If the journal’s policy requires sharing code, it may be helpful to provide the code used for the Principal Component Analysis (PCA) to enhance transparency and reproducibility.

The code is all available publicly via OSF. <https://github.com/dfancourt/socialdeficits>

Reviewer #2 (Remarks to the Author):

The paper examines the relationship between social deficits and ageing, using perceived and physiological age as outcome measures. While the study benefits from a large dataset and the inclusion of multiple biological and self-reported indicators, there are substantial conceptual and methodological weaknesses that limit the validity of its conclusions. Key concerns include ambiguous causal claims, inconsistent variable classification, methodological weaknesses in subjective age calculation; questionable categorization of continuous variables; unclear mechanistic claims. Given these issues, the interpretability and robustness of the findings remain questionable.

We are grateful to the reviewer for their comments on this manuscript and they raise some important points. We feel it is important to clarify that our aim was not produce an externally-validated or central physiological age, but to produce a useful summary measure of physiological ageing in the ELSA cohort. The method we use has already been published and validated elsewhere, as we reference, and we are able to demonstrate that many of the issues the reviewer raises below are actually not problems in our index (e.g. linearity and correlations with chronological age). We have also provided much

more information within the limitations section of the paper and undertaken multiple new analyses in line with the reviewer's recommendations. We acknowledge that there are many different ways to derive ageing indices, but we still believe that our method has value. We have articulated this value and why these analyses remain important in our paper and the responses below.

- Abstract & throughout the ms: The manuscript states that the study provides insights into ageing-related mechanistic processes, yet the analysis primarily examines associations between perceived age, physiological age, and social support. There is no clear justification for how these associations inform underlying mechanisms of ageing. Correlational findings do not establish causal pathways or explain biological or psychological processes that drive ageing. Without a theoretical or empirical framework linking these variables to mechanistic ageing processes, the claim remains unsupported. The authors should either (1) clearly articulate how their findings contribute to mechanistic understanding or (2) revise their conclusions to reflect that the study identifies associations rather than mechanisms.

We have now removed all assertions that we have identified new mechanisms. Instead, we have focused the paper much more clearly on demonstrating that there is a relationship between social deficits and the combined index of these phenotypic markers that considers their predictive potential for ageing. Given this, we have proposed a hypothesis that accelerated physiological age is a potential causal mechanism by which social deficits could influence age-related morbidity and mortality. While this study (which focused only on cross-sectional and some initial longitudinal data over four years of follow-up) cannot make causal claims nor confirm such a mechanism, we put forward the hypothesis here as the basis for future research.

- Abstract & throughout the ms: The term "social deficits" to describe lack of social support, loneliness, and related constructs is problematic, as it implies a deficiency-based or pathological framing of social experiences. These are context-dependent and multidimensional and should not be reduced to a deficit model without justification. The authors should use more precise terms, such as "social support," "social isolation," or "social networks," to align with established terminology.

Across the paper, each term is used individually in relation to our outcomes. We do not create any "index" that represents an overarching concept of "social deficits". Instead, we use the term as to refer to a lack of social support, social isolation, limited social networks etc. This is in line with the most recent theoretical thinking in the field e.g. Holt-Lunstad 2022 <https://doi.org/10.1146/annurev-publhealth-052020-110732>. Each time we refer to specific types of social deficits we use their correct terminology, e.g. "social support". However, we acknowledge that the term "deficits" can be problematic in suggesting normative levels. So we have now rephrased this throughout as "weak social connections".

- The manuscript contains several typographical errors, such as "metaboloic" (Abstract). A thorough proofreading is necessary to ensure accuracy and clarity throughout the text.

This has now been done, thank you.

Introduction

- The claim that "social deficits increase with age" oversimplifies social changes in later life. Research, including Socioemotional Selectivity Theory (SST), shows that older adults prioritize quality over quantity in relationships rather than experiencing a uniform decline. Reduced social contact does not necessarily indicate a deficit. The authors should consult existing theories on social support across the lifespan and revise this statement to reflect these nuanced patterns.

We are grateful for this point, which we agree is important. We now clarify in the introduction: "Multi-national research demonstrates that social connections change over the life-course, with individuals prioritising certain types of social goals at different life points, such as focusing more on quality rather than quantity of social connections as they age. However, there are certain patterns that are well-evidenced and recognised as indicative not just of adaptive or compensatory patterns but instead of weaknesses in social connections that can be detrimental to health." As described above, we have changed our language from deficits to weak connections.

- The introduction is lengthy in parts, particularly in its deep dive into the history of biological ageing, yet it lacks a clear trajectory toward the study's main research question as there is no link to established physiological. As a result, the reader is left uncertain about the study's focus and contribution. The authors should streamline the intro, ensuring that each section directly builds toward the study's aims.

We have now rewritten the introduction so that it is much more concise and provides a clear flow for our arguments.

- Again, one of the biggest issues is the misuse of causal language in an observational study, such as stating that "biological age has been strongly influenced by social deficits." This implies a directional effect that has not been

empirically established. The authors should use association-based language (e.g., "is correlated with") unless they provide strong theoretical and methodological evidence to support causal claims. Ensuring precise language is essential for clarity and validity.

We have now removed all statements that we can identify that appear too strongly causal.

- The term "perceived age" is too ambiguous, as it is often used to describe how others perceive a person's age, rather than an individual's own subjective experience of ageing. If the authors are referring to how old participants feel rather than how old they appear to others, the more precise term would be "subjective age" or "felt age". Clarifying this terminology would prevent misinterpretation and align with established literature on self-perceptions of ageing.

We have now made this change throughout.

- Referring to perceived age as a biomarker is conceptually problematic. While perceived age can be an important predictor of health outcomes and mortality, it does not meet the standard definition of a biomarker, which refers to objective, biological measures. Perceived age is a subjective or socially influenced measure rather than a direct biological marker of ageing. The authors should reconsider this classification.

This was actually a typo. We have now changed this term to "marker" instead.

- The claim that "perceived age provides a mental map of one's remaining life expectancy" is overstated. While subjective age correlates with subjective life expectancy, they are distinct constructs and do not perfectly overlap. The authors should revise this statement to reflect their partial but imperfect association.

We have now removed this statement as part of our re-writing of the introduction.

METHODOLOGY

- A major methodological issue in the paper is the categorization of continuous variables into extreme groups (e.g., lowest vs. quartiles). Dichotomizing or grouping continuous measures, such as depression (yes, no); activity (yes no), etc. , leads to a significant loss of information, reduces statistical power, and may introduce artificial thresholds that do not reflect the underlying distribution of the data. If the variable is inherently continuous, the authors should avoid arbitrary splits and instead model it as a continuous predictor to retain the full variability and improve analytical precision.

We have now re-run the main analyses using linear variables for all variables where this is possible, which shows no material change in findings (see Supplementary Figure 12). We discuss the findings in the paper. This approach is not possible for the exposure, as we use a doubly-robust estimation approach, which is a statistical method for estimating causal effects in different treatment groups. As such, it involves treatment assignment and does not work with linear variables. Using this approach is important as it enables a more rigorous approach to confounders to be taken above linear regression models, and it also responds to calls within social epidemiology as part of work on the potential outcomes framework and target trial emulation for clearer specification of target populations for public health intervention (which we now describe and cite in the paper). Naturally different cut-offs could be applied, but there are not well-recognised standards or cut-offs in most social connections measures. So, a quartile approach provides a good compromise between isolating individuals who are at the lower end of the scale in their scores but also maintaining sufficient statistical power to identify differences. Please note we already provide sensitivity analyses looking at different thresholds. We argue that the new analyses using linear covariates do enhance the paper and we have now provided much more discussion justifying the binary exposure while also considering the limitations within the methods and discussion.

- The term "perceived age acceleration" is misleading and conceptually incorrect in this context. The data only capture a single time point, meaning it is impossible to determine whether participants have experienced a change/acceleration in the discrepancy between subjective age and chronological age over time. Acceleration implies an increasing discrepancy over multiple time points, which is how the term is used in the existing literature. If the difference between subjective and chronological age is stable, it does not reflect acceleration. It's possible that some people have always felt this way and this is a trait with zero acceleration. The authors should either revise the terminology to avoid implying change over time or incorporate longitudinal data if acceleration is the intended construct. Same is true for the term "physiological age acceleration". It is misleading and conceptually incorrect as the data only capture a single time point, meaning it is impossible to determine whether participants have experienced a change/acceleration in the discrepancy between physiological age and chronological age.

"Accelerated physiological ageing" in the biological ageing literature refers to a chronological age higher than a biological age and in this context does not necessarily imply a longitudinal change and is a frequently-used term (e.g.

<https://doi.org/10.1186/s12889-024-17960-w>, <https://doi.org/10.1111/jgs.18611>, <https://doi.org/10.1186/s12889-025->

22053-3.) We have now added additional text into the methods section with these references as justification for using this language but caveating the fact that the term “accelerated physiological ageing” does not necessarily imply a longitudinal change.

However, we acknowledge that subjective age may be more accurately described as “older subjective age”, so we have made this change throughout the manuscript.

- The calculation of subjective/felt age as “perceived age minus chronological age” does not appropriately account for age-related scaling differences. A one-year discrepancy is more meaningful for a 20-year-old than for a 90-year-old. To address this, most studies compute a proportional discrepancy by dividing the subjective age difference by chronological age (i.e., $(\text{Subjective Age} - \text{Chronological Age}) / \text{Chronological Age}$), which better accounts for relative differences across the lifespan. The authors should consider aligning their approach with this established method (Rubin and Berntsen 2006; Stephan et al. 2015) to improve comparability with existing literature. Also, proportional discrepancy scores three standard deviations above or below the mean need to be considered outliers (Stephan et al. 2015; Weiss and Lang 2012) and replaced.

This is an important point if looking at the whole life course. But our sample only looks at adults over the age of 50, where the relative difference in a year is less strong. Nonetheless, we appreciate the suggestion of an alternative approach. We have now added a sensitivity analysis that uses the alternative approach suggested by the reviewer. (There were no outliers beyond 3 SD of the mean.) Notably, the results are materially unaffected, showing exactly the same patterns as in our original analyses (Supplementary Figure 10). So we have elected to keep our original analyses as the main analyses given they are more commonly used in analyses with older adults but we discuss the similarity of the findings using the alternative computation within the paper to highlight that our findings are not just an artefact of our chosen algorithm.

- The approach used to derive the physiological age acceleration measure raises several methodological concerns:
 - o Heterogeneous Temporal Dynamics: The included physiological systems (cardiovascular, metabolic, cognitive – is this even physiological?), etc.) exhibit different rates of change across the lifespan. Some measures (e.g., blood pressure, grip strength) might show a more linear age-related decline, while others (e.g., cognitive function) might exhibit nonlinear trajectories and are influenced by cumulative exposures. Combining these without considering their distinct temporal orders may lead to severe misinterpretation of ageing processes.

As part of biomarker selection, we visually investigated the linearity of associations between physiological measures and chronological ageing using local polynomial regression. We have now added these plots to the Supplementary materials. This line fitting technique is highly prone to overfitting the data, and therefore some deviation from linearity is to be expected. Nonetheless, the trends we observed for the age range under consideration were overwhelmingly linear for all the measures included in physiological age. As such, non-linearity of associations between biomarkers and physiological age is unlikely to pose an issue in this particular study population. We have now included a new figure (Supplementary Figure 1) plotting standardised biomarkers against chronological age using polynomial regression to illustrate this.

- o Causality and Interdependencies: The included indicators are not independent—some serve as causes, others as consequences of physiological ageing. For example, metabolic dysfunction may contribute to cardiovascular deterioration,

which in turn affects cognition. Ignoring these interdependencies may distort the meaning of the derived physiological age acceleration scores.

There is no supposition in the derivation of physiological or biological ages that the indicators should be independent. Indeed, this is not possible or logical if the underlying theory is that ageing is driven by a set of central processes giving rise to the biomarker changes that we use to produce physiological age. Each indicator will serve as both a consequence of ageing processes, and a cause of subsequent biomarker changes, and it is not possible to distinguish between these or select a set of biomarkers that are strictly consequences but not causes of other indicator changes. So our analysis makes no such supposition.

o Principal Component Analysis (PCA) seems inappropriate → PCA assumes that the included variables represent independent dimensions of variation, but ageing is a highly correlated, hierarchical process. Using PCA for dimensionality reduction may mix cause and consequence, and the first component may capture variance unrelated to ageing per se. Alternative methods such as latent variable modelling (e.g., structural equation modelling) could better account for hierarchical dependencies. The authors include chronological age in the PCA, predict component scores, examine correlations, and then remove chronological age in a second PCA to compute physiological age. This approach introduces two key issues. First, including chronological age in the first PCA forces the first component to capture chronological age variance, making it unclear whether the component represents ageing-related physiological changes or just chronological age itself. Second, By running PCA again without chronological age, the method disconnects the derived score from the variable initially used to define biomarker importance. This creates an inconsistency in how ageing is defined and how the final score is constructed.

The PCA method is among the most commonly used methods of biological age derivation (e.g., <https://pubmed.ncbi.nlm.nih.gov/12634284/>, <https://pubmed.ncbi.nlm.nih.gov/19940465/>, <https://pubmed.ncbi.nlm.nih.gov/19940465/>, <https://pubmed.ncbi.nlm.nih.gov/22433233/>, <https://pubmed.ncbi.nlm.nih.gov/24522464/>, <https://pubmed.ncbi.nlm.nih.gov/24659482/>, <https://pubmed.ncbi.nlm.nih.gov/3226152/>, <https://pubmed.ncbi.nlm.nih.gov/18597867/>, <https://pubmed.ncbi.nlm.nih.gov/18840798/>, <https://pubmed.ncbi.nlm.nih.gov/8803500/>, <https://doi.org/10.1186/s12889-024-17960-w>). It is also amongst the most intuitive.

We agree with the reviewer (and we acknowledge in the paper) that there are limitations to the PCA method, though we once again emphasise that our aim was not produce an externally-validated or central physiological age, but to produce a useful summary measure of physiological ageing in the ELSA cohort.

Though we agree with the reviewer that SEM methods are useful for more nuanced selection of indicators, in the present study, our interest is primarily in developing an easily interpretable summary measure of physiological age that can be compared with chronological age. While we acknowledge that we lose some nuance in this approach, it is important to note that SEM methods produce estimates that are not straightforward to compare with chronological age (<http://dx.doi.org/10.1093/pnasnexus/pgac135>, <https://www.tandfonline.com/doi/full/10.2147/CIA.S134921#d1e2162>). So, for our study aims, the PCA method is more practical and meets our needs. We nonetheless agree with the reviewer that the limitations of the method need to be better acknowledged in the limitations section, and have revised the manuscript to reflect this, with an expanded discussion on this point in the limitations section.

o The calculation of physiological age acceleration (physiological age – chronological age) assumes a linear relationship between physiological and chronological ageing. However, this assumption is problematic given the nonlinear nature of biological ageing. Some physiological systems may decline early (e.g., lung function), while others remain stable longer (e.g., cognitive function), making a single age difference score potentially misleading. Physiological age is linearly associated with chronological age for the age range under study, and the indicators included are linearly related to age, so this is not a consideration for our study. We have now added a plot the Supplementary materials (Supplementary Figure 2) fitted using local polynomial regression that shows the linearity of the relationship.

o While the authors state that the index is associated with functional limitations and chronic conditions, they should provide rigorous validation analyses, including comparison with existing biological age models (e.g., epigenetic clocks, telomere length, brain age, skin age, etc. etc.).

On a biological level we disagree with this suggestion. It is not typical to see validations of physiological ageing indices against biological age models like epigenetic clocks as they measure different aspects of ageing (downstream phenotypic measures as incorporated in our index vs upstream telomere or epigenetic processes within the 'omics cascade). It is widely recognised that downstream vs upstream biological measures do not typically correlate as they represent very different biological processes (e.g. <https://elifesciences.org/articles/59479.pdf>). To clarify, our physiological ageing index is not meant to be comparable to telomere data or epigenetic clocks; it is meant to be a summary measure of phenotypic aspects of physiological ageing in this population. (As an aside, methylation-based measures are not even included in ELSA). Instead, physiological ageing measures based on clinical indicators are useful in settings where these data are not available and do have a strong precedent, as outlined in the citations already provided above.

o Biomarkers were pre-selected based on their correlation with chronological age ($r \geq 0.10$), ensuring that the final physiological age score is artificially correlated with age. This approach excludes biomarkers that may reflect physiological ageing independent of chronological age, potentially skewing results. It is possible that an indicator is not highly correlated at one point in the lifespan but yet predictive for mortality outcomes. The chosen correlation cutoffs ($r \geq 0.10$ for selection, etc.) appear arbitrary and lack justification. A weak correlation with age ($r \geq 0.10$) does not ensure that a biomarker reliably captures physiological ageing. A moderate correlation ($r \geq 0.30$) with the PCA score does not confirm biological relevance, as PCA prioritizes variance, not ageing-specific processes.

We have chosen a summary measure of physiological age appropriate for our study question and have followed the PCA protocol applied in other studies. Other methods of biomarker selection and biological age derivation such as SEM might better account for complex interrelationships between biomarkers. Nevertheless, our aim was to produce an easily interpretable summary measure of physiological ageing that is straightforward to compare with chronological age, leading us to use the well-established PCA method. We now add new text acknowledging the limitations of the method within in the limitations section.

o Most importantly, PCA assumes that all retained physiological measure the same underlying construct, ignoring the heterogeneous and multi-system nature of ageing. Ageing affects different systems (e.g., metabolic, immune, cardiovascular) at different rates; reducing it to a single component oversimplifies complex biological processes and ignoring the heterogeneous nature of ageing across different organ systems.

We agree that work on organ-specific ageing is very exciting, but it asks a fundamentally different research question to what this paper set out to achieve. There is extensive research that focuses on overarching biological indices of physiological age, just like ours, which we now cite in the paper (e.g., <https://pubmed.ncbi.nlm.nih.gov/12634284/>, <https://pubmed.ncbi.nlm.nih.gov/19940465/>, <https://pubmed.ncbi.nlm.nih.gov/19940465/>, <https://pubmed.ncbi.nlm.nih.gov/22433233/>, <https://pubmed.ncbi.nlm.nih.gov/24522464/>, <https://pubmed.ncbi.nlm.nih.gov/24659482/>, <https://pubmed.ncbi.nlm.nih.gov/3226152/>, <https://pubmed.ncbi.nlm.nih.gov/18597867/>, <https://pubmed.ncbi.nlm.nih.gov/18840798/>, <https://pubmed.ncbi.nlm.nih.gov/8803500/>, <https://doi.org/10.1186/s12889-024-17960-w>,). This work is critical to build an understanding about overall associations with physiological ageing before more organ-specific hypotheses could be generated. So we focused on this within our paper, but we have now made a recommendation that future research explores organ-specific indices within our discussion section.

- The use of doubly robust estimation with the IPWRA estimator in a cross-sectional study raises some methodological concerns. First, the lack of temporal ordering prevents causal inference, as exposure and outcome are measured simultaneously. Second, unobserved confounders may bias the results. Third, inverse probability weights (IPWs) are more appropriate for longitudinal studies, where they adjust for selection bias over time; their use in a cross-sectional setting is less effective. Fourth, many covariates, including sex and health behaviors, vary over time, but the analysis relies on a single time-point measurement. Finally, the cross-sectional design limits the ability to conduct sensitivity analyses to assess time-dependent confounding or feedback effects.

Doubly robust estimations naturally have limitations, but they also have many strengths. They maintain unbiased estimates even if one model is misspecified. They can be more efficient compared to traditional IPW estimation. They allow greater flexibility in the choice of models, allowing for better control of confounding. They simultaneously provide relative and absolute effects. There are ample examples of their usage in cross-sectional studies (e.g. <https://doi.org/10.1093/biomet/ass013>), with comparative studies suggesting very similar estimates from approaches such as IPW and adjusted regression models (e.g. 10.1097/EDE.0b013e3181f57571).

Regarding the cross-sectional issue, we do want to flag that we DID have a longitudinal data component, not only looking cross-sectionally but also with data four years later. Naturally, incorporating additional longitudinal data will be an important part of future analyses to build on the research presented here, but our analyses provide important first data on this. We have now provided additional discussion of the limitations, discussing the issues raised by the reviewer here, and proposed how the work can be taken forwards in the limitations section.

- The classification of variables as confounders, mediators, or outcomes appears arbitrary and lacks clear conceptual and empirical justification. For example, depression could plausibly act as a mediator, confounder, or bidirectional factor in the relationship between social connections and ageing, depending on the theoretical framework. The decision to include certain variables in the exposure vs. outcome confounder models seems inconsistent and requires a stronger theoretical rationale.

We have now added in substantial additional text and references to justify our theoretical rationale for how we are treating confounders within the methods section.

- Table 1 lists "No. of chronic conditions" as "Yes/No," which seems wrong—a count variable cannot be binary. The authors should clarify whether they measured the number of conditions or the presence of any condition and adjust the categorization accordingly.

This was a typo and has now been corrected.

- The manuscript lacks a full intercorrelation table for all study variables, except for social support. Given the potential for multicollinearity, particularly among ageing-related psychological measures, the authors should provide a full correlation matrix to assess the extent of redundancy and ensure the validity of their models.

We have now included an additional correlation matrix for all variables in the study demonstrating no multicollinearity or variable redundancy. This is in Supplementary Figure 4.

- The pyramid structure used for the social connection model implies an implicit hierarchy or ordering of social relationships that lacks empirical justification. If the model suggests qualitative differences between levels, the authors should provide empirical support for this structure. Otherwise, an alternative visualization that does not imply a strict ranking may be more appropriate.

The pyramid figure is based on existing theory e.g. Holt-lunstad & Steptoe 2022, as structural factors are thought to underpin functional and quality factors (see figure below from the paper mentioned for reference). For example, an individual cannot have poor quality social relationships if they are entirely socially isolated. We have now clarified that our pyramidal structure is based on pre-existing theory in the methods and provided an accompanying reference.

- Resolution of figures is poor.
This has now been updated.

DISCUSSION:

- Additionally, the literature review is highly selective, overlooking key studies in ageing, social support, and biological ageing. The authors should engage more comprehensively with existing research to provide a balanced and accurate foundation for their work.

We have now added in substantial amounts of new literature to the paper (over 20 new references).

- The claim that the study provides "novel insight into a plausible mechanism" linking social deficits to ageing is overstated and inaccurate. Demonstrating an association between social support and a composite ageing index does not establish a mechanism, as mechanisms require causal pathways explaining how social factors biologically influence ageing. The authors should clarify their contribution—if they are identifying correlations, they should not claim mechanistic insight. Additionally, the novelty of this finding is questionable, given prior research on social determinants of (subjective and biological) ageing. Similarly, the claim that these findings "provide insight into how social deficits influence age-related pathology" is overstated. Additionally, the statement that "objective biological and functional processes may be more important mechanistically than broader processes related to subjective perceptions" is unclear and lacks empirical justification. The authors should clarify their reasoning, define what is meant by "broader processes," and avoid mechanistic claims.

We have now removed claims that our study shows new mechanisms as outlined above. We have nuanced statements about the relevance for age-related pathology research too, such as changing the sentence mentioned to "these findings may have a relevance for research focused on social deficits and age-related pathology", and we have then turned our assertions into further hypotheses to be tested in future studies.

Reviewer #2 (Remarks on code availability):

Did not see the code. The code is all available publicly via OSF. <https://github.com/dfancourt/socialdeficits>

Reviewer #3 (Remarks to the Author):

Authors used ELSA data to evaluate social connections (structural, functional, and quality) on perceived and physiological aging. Overall, the study was well written and contributes to the literature in a meaningful way. However, I have a few recommendations to strengthen the contribution and justification for the design of the study.

We are pleased that the reviewer is positive about the paper.

Introduction:

I would recommend rewording the lines 44-54 as it makes it seem that physiological age is new – even though it precedes the development of epigenetic clocks and metabolic clocks. I would perhaps also focus on the fact these types of clocks were not trained on prediction of outcomes but rather incorporate physiological aging processes themselves (unlike epigenetic clocks that were trained on outcomes).

Thank you for this point. This clarification has now been made.

I am uncertain about the current motivation for perceived age and physiological age. It seems like two different papers to me. I would recommend some additional incorporation that may answer is one related to health better than the other? How are they related to each other in other studies since it has low correlation here? Why are both necessary to understand the mechanistic processes? Giving further motivation here would help to strengthen the contribution of the paper and highlight why we need to consider perceived (even if you the authors do not find as robust findings).
In line with Reviewer 2's comments, we have now rewritten the introduction. Our motivation for including both is to ascertain how much results are due to subjective perceptions vs objective physiological markers of ageing.

Methods:

Can the authors motivate the addition of cognitive functioning as a component of physiological age, as cognition is not purely a physiological process but a combination of cognitive development and pathology. To justify this use, I might also recommend the authors do a sensitive check without cognitive functioning to keep in line with prior iterations. I do recognize that it is unlikely to change the results as cognitive functioning often runs parallel to the other components, but conceptual clarity would be helpful here.

We agree that cognitive function is not purely physiological and that including it in a physiological age measure introduces conceptual ambiguity. Our rationale for its inclusion was that cognitive decline often parallels physiological deterioration in ageing and may share underlying biological pathways (e.g., inflammation, vascular health). That said, we recognise the importance of conceptual clarity and have conducted a sensitivity analysis excluding cognitive function from the physiological age composite. The results remain substantively unchanged.

Discussion:

What does the role of mortality selection have on these models? Is this a limitation to consider? I appreciate the 4 years follow up but I am still wondering about sample selection and survival and important processes to consider.

We agree that the people with the worst social deficits and fastest physiological ageing may be less likely to survive to be included in the study. For the cross-sectional analysis, this would tend to bias associations toward the null. However, this would mean that if anything, the results underestimate the association between social deficits and physiological ageing. For the longitudinal analysis, survival could technically be considered a collider and potentially therefore induce a spurious association. Nonetheless, the fact that longitudinal results are consistent with cross-sectional findings suggests this was unlikely to be an issue. We now add a new note on this into the limitations section.

Could the authors also add that people with health conditions may also have more complications with social participation, which itself may lead to faster aging? I realize that the living alone and living with someone may partly address this concern (especially with severe health conditions) but feel it may be an important note to consider for future work (or even a potential sensitivity check).

We have now provided an expanded paragraph in the discussion (paragraph 2) where we discuss bidirectionality in much greater detail and explain that while we cannot make causal claims from this data, we do put forward some hypotheses to be tested in future causal studies.

Minor comments:

You write the first aging clocks in the introduction. I would reword to state the first-generation epigenetic aging clocks were trained on chronological age. While not called aging clocks per se, other algorithms like PhenoAge served a similar purpose and may confuse readers.

We have now revised the introduction in line with comments from other reviewers so this has been removed.

Please specify if sample was community dwelling adults. If so, that may also be a limitation to consider. But I am uncertain of the commonality of nursing homes in the UK context.

Our sample focused on community-dwelling adults, so it does not incorporate adults living in care homes. We have now listed this as a limitation, although we have highlighted that only around 2.5% of adults over the age of 65 in the UK live in care homes.

RESPONSE TO REVIEWERS

Reviewer #2 (Remarks to the Author):

I appreciate the authors responsiveness. The authors acknowledge the concerns raised. However, the manuscript has the potential for deeper improvement.

We are grateful that the reviewer appreciates all of our updates to the paper, and we're delighted that the other 3 reviewers are satisfied with the paper now and have approved it for publication. We have made changes for the reviewer's remaining suggestions below.

Page 2; first paragraph: The revised framing improves clarity but still misrepresents the state of the literature. The claim that physiological aging indices are "providing new avenues" overstates their novelty. These indices have been developed and used for over a decade, and many researchers explicitly avoid combining heterogeneous systems (e.g., cardiovascular, inflammatory, functional) into a single composite due to their distinct biological trajectories and limited conceptual coherence. Moreover, the field has moved toward more mechanistically grounded approaches such as epigenetic clocks, which may reflect upstream aging processes more precisely. The revised paragraph would benefit from a more balanced acknowledgment that physiological indices are not conceptually new, and their aggregation remains contested. This would also temper the overstated novelty in linking them to social variables—something that has been explored, if not always in aggregated form.

We have now rephrased this as "alternative avenues", which illustrates that this is a different type of research from that carried out previously, but removes the potential for overstated novelty. That said, we do want to stress that aggregate physiological indices are novel in relation to social variables and there is increasing acceptance of their importance in relation to complex exposures (e.g. <https://www.nature.com/articles/s41591-025-03808-2>).

Page 2; second paragraph: The claim that subjective and physiological age provide "complementary insight into mechanistic processes" remains rather speculative without theoretical or empirical justification. A more cautious and theoretically grounded framing is needed to clarify the study's actual contribution.

We have provided substantial additional theoretical framing in our introduction for why subjective and physiological age provide potential mechanistic insight. In response to previous comments, we clarified in this paragraph that we are not testing the mechanism specifically, but taking a first step towards such a test by looking at the association between subjective and physiological age and weak social connections. We have been cautious in providing this explanation and in our discussion in interpreting our findings. But we do believe it is important to keep this sentence in the introduction – it explains clearly and concisely why this work is necessary and where it can lead, which is important for readers to understand the trajectory of this line of research.

The authors acknowledge concerns about heterogeneous aging dynamics but largely dismiss them by relying on visual inspection using local polynomial regression. While the added plots in the Supplementary Materials are appreciated, the justification remains insufficient. Local polynomial smoothing is, as the authors note, prone to overfitting and does not provide formal tests of nonlinearity. More importantly, visual linearity within a constrained age range does not negate the conceptual issue: the biomarkers reflect distinct physiological systems with inherently different temporal trajectories and exposures. Simply showing approximate linearity does not resolve the concern about combining them into a single metric.

The reviewer makes two points here, relating to whether it is statistically appropriate and conceptually appropriate to include different physiological systems within a single ageing metric. **Statistically**, we have now undertaken even more checks on this issue. We have formally assessed whether age-biomarker associations were linear by fitting restricted cubic splines (knots at the 5th, 35th, 65th, and 95th percentiles of age in the analytic sample). Likelihood ratio tests of nonlinearity were statistically significant for three of eight biomarkers, reflecting departures from a strictly linear relationship with age. However, consistent with the local polynomial plots, spline-based plots showed that in the analytic age range all biomarkers increased monotonically with age. These findings indicate that while the large sample size makes even small deviations from linearity detectable, the functional forms are close to linear and monotonic. This is important because the PCA-based physiological age index does not require perfectly linear relationships, only that biomarkers show consistent age-related change. Monotonicity ensures that individual ordering is preserved, so the PCA is robust to these minor nonlinearities. We have now replaced local polynomial plots with spline-based plots in the

supplementary materials, and added a supplementary table with p-values for formal tests of linearity. **Conceptually**, there is substantial precedent for indices such as the one presented here, which we already cite as support for the approach presented here. We have provided an extended list here, including a recent paper from Nature Medicine, to show that such indices are being widely used:

<https://pubmed.ncbi.nlm.nih.gov/12634284/>, <https://pubmed.ncbi.nlm.nih.gov/19940465/>,
<https://pubmed.ncbi.nlm.nih.gov/19940465/>, <https://pubmed.ncbi.nlm.nih.gov/22433233/>,
<https://pubmed.ncbi.nlm.nih.gov/24522464/>, <https://pubmed.ncbi.nlm.nih.gov/24659482/>,
<https://pubmed.ncbi.nlm.nih.gov/3226152/>, <https://pubmed.ncbi.nlm.nih.gov/18597867/>,
<https://pubmed.ncbi.nlm.nih.gov/18840798/>, <https://pubmed.ncbi.nlm.nih.gov/8803500/>,
<https://doi.org/10.1186/s12889-024-17960-w>, <https://www.nature.com/articles/s41591-025-03808-2>

The inclusion of memory as part of the physiological aging index remains conceptually problematic. While cognitive decline is age-related, it is not equivalent to brain age, nor is it typically considered a direct marker of physiological aging. Brain age clocks rely on structural and functional neuroimaging or molecular data—distinct from behavioral measures of memory performance. Equating memory scores with brain age in their line of reasoning conflates separate constructs. Although the authors conducted sensitivity analyses excluding cognition, the rationale for its inclusion still lacks theoretical clarity. The manuscript should more explicitly acknowledge these conceptual distinctions and avoid overstating the relevance of memory as a physiological indicator. Relatedly, the term “memory” is too general to accurately describe the cognitive measures used in the index. Tasks assess specific aspects of episodic memory. Given the differences between memory domains—and their distinct aging trajectories—the manuscript should specify that the measure reflects episodic memory performance.

We appreciate that the reviewer is still concerned about the inclusion of memory in the index, and understand the rationale given for its exclusion. In response to this comment, we have now taken memory out of the index. The tables and figures are now updated to reflect this. The new results are materially very similar to those before, so our conclusions are not affected.

Figures 3 & 4: The x-axis label needs to be updated to the new terminology- “differences in perceived age (years)”

This has now been done.

Page 12, 2nd paragraph: The revised section introduces useful nuance by acknowledging bidirectionality and the limits of causal inference, but it still leans too heavily on speculative interpretation. Moreover, positing this as a “potential causal mechanism” risks overstating the implications of what remains correlational evidence—particularly given the composite nature of the physiological index and the lack of temporal resolution. I recommend more cautious wording that emphasizes hypothesis generation rather than mechanistic interpretation.

We respectfully note that this section is not presented as an interpretation of our findings, but rather as a hypothesis for future research. We have already carefully revised the text to state explicitly that this is hypothesis generation and not an interpretation of our results, for example: “While this study (which focused only on cross-sectional and some initial longitudinal data) cannot make causal claims nor confirm such a mechanism, we put forward the hypothesis here as the basis for future research.” We also clarify that this is hypothesis generation rather than mechanistic interpretation in our discussion section. We hope that this cautious wording is now clear to the reader.